# “Found in Translation”: An Evolutionary Framework for Auditory–Visual Relationships

**DOI:** 10.3390/e24121706

**Published:** 2022-11-22

**Authors:** Ana Rodrigues, Bruna Sousa, Amílcar Cardoso, Penousal Machado

**Affiliations:** 1Department of Informatics Engineering, Centre for Informatics and Systems of the University of Coimbra, University of Coimbra, 3004-531 Coimbra, Portugal; 2Intelligent Systems Associate Laboratory (LASI), University of Minho, 4800-058 Guimarães, Portugal

**Keywords:** auditory–visual associations, computational creativity, genetic programming, emotions, perception, abstract art, music generation, image generation

## Abstract

The development of computational artifacts to study cross-modal associations has been a growing research topic, as they allow new degrees of abstraction. In this context, we propose a novel approach to the computational exploration of relationships between music and abstract images, grounded by findings from cognitive sciences (emotion and perception). Due to the problem’s high-level nature, we rely on evolutionary programming techniques to evolve this audio–visual dialogue. To articulate the complexity of the problem, we develop a framework with four modules: (i) vocabulary set, (ii) music generator, (iii) image generator, and (iv) evolutionary engine. We test our approach by evolving a given music set to a corresponding set of images, steered by the expression of four emotions (angry, calm, happy, sad). Then, we perform preliminary user tests to evaluate if the user’s perception is consistent with the system’s expression. Results suggest an agreement between the user’s emotional perception of the music–image pairs and the system outcomes, favoring the integration of cognitive science knowledge. We also discuss the benefit of employing evolutionary strategies, such as genetic programming on multi-modal problems of a creative nature. Overall, this research contributes to a better understanding of the foundations of auditory–visual associations mediated by emotions and perception.

## 1. Introduction

Abstract languages, such as music or images, can be complex to understand. Sometimes, it may even take several years of training before humans can communicate through them. However, when properly conveyed, they can be rich sources for communication and hold the capacity to achieve more expressive and comprehensive communication than verbal languages.

Understanding how to express a piece of music or image is crucial for developing multi-modal artifacts. However, the expression of music or image is not a mere sum of its characteristics. It depends on many factors, such as culture, personal interests, physical aspects, and how these properties are processed at cognitive levels. Examples of perceptually based tools that relate auditory and visual dimensions can be found in references [1,2]. However, in these works, a limited number of perceptual features are explored, and the established mappings are static, that is, they do not adapt depending on the context. Conversely, the few works that have considered evolving the associations between music and image [3,4] do not show evidence of relying on the findings of cognitive sciences to bridge these two creative domains.

Crossing perceptual and emotional findings with auditory and visual characteristics may bring additional value to the expression of cross-modal associations of a creative nature. As such, we rely on music and image elements, emotions, and perception studies to draw a dataset of the most perceptually relevant properties for our study. Our choice of working with abstract images is grounded on a study by Melcher et al. [5], where they suggest that the perception of abstract artworks can be used efficiently in the representation of emotions. We note that we are working toward emotion perception (EP), which is not the same as emotion induction [6,7].

Moreover, the speed and modularity of current technology adds new forms of creation, together with an ability to interpret and evolve abstract concepts with certain degrees of autonomy [8,9,10,11]. In turn, this expressive capacity of computers opens doors to the development and exploration of creative languages, such as image and music. Driven by the computational potential of evolutionary processes, we propose a novel approach to the development of a creative tool capable of bridging the perceptual qualities of visual and auditory languages (see Figure 1). During the course of this work, on the one hand, we handle the challenge of an abstract representation and transposition of knowledge between multiple domains. On the other hand, we handle the computational expression, that is, the challenge of balancing freedom with rigid rules, a topic long discussed in the field of CC (computational creativity) [9].) and the evolution of these relationships.

To manage the complexity of the problem, we propose a novel framework to develop and explore a set of cross-modal associations between music and image. Hence, our framework is divided into four main modules: (i) a vocabulary + relationship dataset, (ii) a music generator, (iii) an image generator, and (iv) an evolutionary engine. This modular approach promotes flexible navigation in a large space of solutions. Each of these modules is detailed in depth in Section 4.

In a general form, the vocabulary allows the transposition of knowledge between domains. It contains a perceptual organization of visual and auditory properties, a set of emotions common to both domains, and a dataset of relationships between these. The generators of music (MG) and image (IG) were built to support the expression of several emotions or sensations dynamically. This includes the definition of grammar rules and an interpreter based on a context (e.g., emotions). We should note that although emotions can drive the generation of music and image, these generators are not limited to it. For proof-of-concept purposes, we focus our study on four emotions: anger, calm, happiness, and sadness [12]. Regarding the EE (evolutionary engine), we propose an evolutionary solution for the adaption and expansion of the audio–visual language. To accomplish this, we use genetic programming techniques, as they are efficient in solving problems of a high-level abstraction and are domain-independent [13].

We demonstrate how the framework can be used through a practical example. To evaluate the system’s expressive performance, we assemble a preliminary test with users to assess the user’s perception of music–image pairs against the system’s evolved outcomes. The results strengthen further developments in the proposed computational approach for the exploration and study of cross-modal associations between creative domains.

The main contribution of this research lies in bringing together diverse research domains, such as music, image, emotion, cognitive sciences and evolutionary computation, to explore the potential of computational tools for abstract auditory–visual associations. In detail, the development of this research adds three benefits: (i) offering a systematic and multidisciplinary approach to bridge the domains of music and image based on a shared vocabulary of perceived emotions, (ii) providing a test hypothesis for a system that allows the exploration of auditory–visual relationships, and (iii) contributing to a better understanding to the transposition of human creative processes in computational approaches.

## 2. Auditory–Visual Associations

Whether we are listening to a music or observing an image, we can acknowledge that the group of elements present in it will influence how we sense the whole [14,15]. In the hope of understanding a domain’s emotional contents and perceptual characteristics, we build a dataset of perceptually relevant associations between visual and auditory domains’ properties.

Looking at the history of audiovisual artworks, the types of associations between these domains fall into one of these three categories, as illustrated in Figure 2:1-1: Here, one property of one domain is mapped into one property of another domain. For example, pitch octave → color hue or timbre → form (shape) (examples from the works of M. Ellen Bute and McLaren, respectively) [16,17].1-N/N-1: Here, several properties of one domain are mapped into a single property of another domain. For example, loudness+pitch → brightness [18].N-M: Here, several properties of one domain are mapped into several properties of another. For example, sound duration+pitch → length+movement direction [17,19].

**Figure 2 entropy-24-01706-f002:**
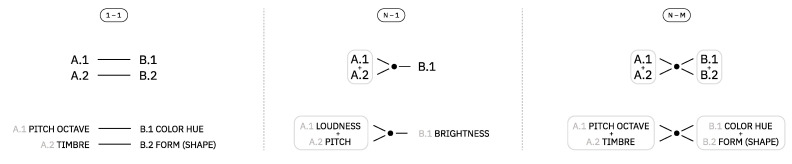
Illustration of the three main types of audio–visual mappings found in the literature.

As for the interpretation of one specific domain set of properties, we found two distinct paths: one where they are expressed in a direct (metric) way, such as visual music examples [20], and one through cognitive findings (such as emotions or perception) [1].

In this study, we build a set of relationships from N-M. This is, we perform an association between several properties of music to several properties of an image (see Section 4.4). Next, we mention examples of direct music–image studies due to their artistic expressivity. However, they are not included in our analysis, as they generally lack empirical testing and are merely founded on each individual’s convictions.

Earlier artists and painters believed that certain combinations of color, form, and movement did emphasize the AV experience. Therefore, they performed several experiments that generally consisted of a direct translation from music to image, i.e., mapping between a set of auditory and visual properties. The Visual Music era began around 1910 with abstract painters such as Wassily Kandinsky. Kandinsky was known for his two-dimensional paintings, where he translated music into abstract images—they were often visually characterized by a series of color combinations and object transformations, such as permutation, expansion and contraction [21]. The will to express music in a time-based sequence of graphic patterns not long after made animated visual music emerge. Several techniques have been approached for this end, ranging from hand-drawn animations in metric paper to creating custom-built instruments, such as the clavilux by Thomas Wilfred, to the sound spectrograph by M. Ellen Bute, and computer programs by John Whitney. By 2000, the Whitney brothers began to use a computer to build algorithms inspired by Pythagoras’ theories of harmony and computationally generated a series of patterned and symmetric animations [22].

Each visual music artist would create an individual abstract language based on their own interpretation of musical concepts to explore this relationship without further empirical experimentation. Some artists, such as Bute or Fischinger, expressed the common desire to expand the simple translation and connect the two domains through perception. However, they did not have the technological means to achieve it at the time. More recently, some authors performed experiments based on this idea of perceptual features—the way we see or hear things through our perception. For instance, Grill and Flexer proposed a sound visualization strategy that “ uses metaphoric sensory properties that are shared between sounds and graphics” [1] to build meaningful associations between auditory and visual dimensions. Specifically, the visual features of shape, stroke, grid position, and color were linked to textural sounds classified into the following perceptual categories: smooth/coarse, tonal/noisy, ordered/chaotic, high/low, and homogeneous/heterogeneous [1].

In empirical studies regarding music and emotions, the most common findings relate high loudness and fast tempo with anger or slow tempo and low loudness with sadness. Positive emotions, such as happiness, have been associated with musical features of fast tempo and major modes, whereas negative emotions have been linked to minor modes. Perceptual harmonic aspects, such as dissonance, have been associated with feelings of sadness, excitement, anger and sadness. In contrast, consonance has been associated with feelings of happiness, calm, and tenderness, among others [23,24].

While we can find many experiments associating musical aspects to emotions, examples relating emotions to visual aspects are more limited to studying color (hue, brightness, and saturation) and shape. From the neuroscience point of view, it is known that the visual system is sensitive to a limited number of visual features. Contours (lines), shapes, colors, depth, texture, movement, and composition are the “building blocks of our perceptual experience” [5,25]. In general, brighter colors are most likely to be associated with positive emotions, whereas darker colors are linked to negative emotions [18,26]. Round shapes, for instance, have been primarily associated with positive emotions, such as calm and pleasure, whereas sharp shapes have been linked to negative emotions, such as fear and anger [25,27] (see Figure 3).

In the following subsection, we provide further detail and a structured dataset containing these types of associations for auditory and visual domains.

### 2.1. A Data Map of Audio–Visual Associations Mediated by Emotions

Evidence on cross-modal associations between music, image, and emotions has emphasized the importance of a multidisciplinary approach to studying the relationships between these domains [5,27].

It has been suggested that certain auditory and visual combinations can arouse specific emotions, meaning that a specific set of auditory and visual properties can be used to represent distinct expressions of emotions. While emotional responses to music have been studied extensively and empirically [15,28], the same study in the visual arts still needs further experiments. In this section, we analyze existing research on the two domains and propose a connection between music and image mediated by emotions.

Constrained by other scientists and artists’ work, we assemble each domain’s properties and propose a dataset to connect image and music through affective findings. To this end, we first went through a process of collecting properties able to represent visual and auditory domains. On top of this representation of languages, we established a map of associations mediated by emotions. Further details regarding the property selection process can be consulted Section 2.2 and Section 4.1).

As it was essential to limit the scope of our experiments to extract significant conclusions, we focused on presenting information on four emotional states: angry, calm, happy, and sad. This set of emotions was chosen for two reasons: the amount of information available, and their position in the circumplex model of emotions [12]. The resulting dataset is presented in Table 1.

Defining the search criteria was fundamental to decide the inclusion of specific literature in this study and to determine whether it met our objective of exploring the possible connections between the two abstract non-verbal domains of images and music and EP. To avoid the repetition of findings, articles containing original experiments were favored, and, as such, a limit on the year of publication was not imposed. We then crossed referenced the articles containing a reference to other articles. Finally, we wrote down the association and reference and verified it. We did not include references with an unconfirmed hypothesis in the present study, as we were looking for experimentally confirmed relationships.

Although we gathered a more extensive range of data on AV properties and emotions, we only present the information of properties when three or more of these emotions are mentioned. Therefore, the following properties were excluded from this dataset following this method:Music: harmonic intervals, melodic motion, vibrato, pause, tempo variation, note density, and rhythmic pattern.Image: regularity, closure, opacity, balance, density, visual distribution, augment/diminish, movement, symmetry, repetition, rotation, cluster, depth, distance, variation, volume, and overlap.

In the following section, we present a methodology and discussion on the selection of perceptually relevant characteristics based on the amount of of associations performed by several authors.

### 2.2. A Selection of Perceptually Relevant Characteristics

Considering the extension of information in the given dataset, we performed a methodological process to extract two types of information: (i) the emotions that are most commonly associated with auditory or visual properties; and (ii) the auditory and visual properties that are most commonly used to represent emotions. Our process of information curation and systematization was inspired by a recent study of Caiola et al. [33].

In the first analysis phase, we started by counting how many times each parameter is associated with a specific emotion. Then, we added the total values of all the parameters present in the dataset for that emotion (see Table 2 for an example of this process). Finally, we repeated this process for each emotion. This sum resulted in Table 3. The table analysis shows that, for example, the emotion “happiness" was associated 192 times with visual parameters and 63 times with musical parameters. Furthermore, we can conclude that “sadness”, “happiness”, “fear” and “anger” are the emotions most commonly associated with visual and musical properties, followed by “tenderness" and “calm". These conclusions provided guidance for the choice of emotions in the present study.

In the second analysis phase, we added the number of times each property is associated with each emotion. As a result, we obtained the total number of times a particular property is associated with the whole set of emotions (see Table 4 for an example of this process).

For the most perceptually appropriate analysis, especially when considering the relation of associations between properties, we normalized the values obtained by dividing them by the sum of all the property values. For example, given that we have 252 associations of the property “color”, and knowing that there were a total of 410 associations of visual properties, we calculated the relevance of this property in the overall context of visual property–emotion associations as follows:

(Color Associations × 100)/Total associations252 × 100/410 = 61.46%

In this process, we could conclude that “Color” corresponds to 61.46% of the total number of visual property–emotion associations.

Based on the analysis of Table 5, we can conclude that the most perceptually relevant visual properties to express emotions are color (brightness, hue, saturation), shape, edge, size and texture; the most perceptually relevant musical properties to express emotions are tempo, loudness, articulation, high-frequency energy and harmony sonance.

The results were similar when comparing the most relevant properties associated with the entire set of emotions with the most relevant ones associated only with the four selected emotions. That is because these four emotions have more associations with properties. Nevertheless, it is essential to highlight some differences: for example, regarding visual properties, “thickness” and “regularity” have higher relevance in the limited group of emotions; regarding musical properties, the differences are more expressive, with properties such as “tone attack” and “articulation variation” gaining more relevance, and “pitch level” losing relevance.

Properties with a lower percentage of associations are not necessarily less relevant. This could mean that these properties are less studied in this area and need further research. However, we worked and based our conclusion on the previously collected data to guarantee scientific support.

## 3. Computational Expression of Music and Image

In the past years, many computational systems have been conceived to generate or achieve similar processes to human individual creativity [3,34]. However, often, they do not focus on aesthetic rules or perception principles to guide IG. As we propose a more substantial inclusion of perceptual qualities in the computational tools that support creativity processes, we briefly revisit some related works that contributed to the progress of areas, such as algorithmic composition, computer art, and CC.

Distinct approaches to creative computational systems have been proposed. Sometimes, their processes involve the combination of “familiar ideas in unfamiliar ways” [3], and, in other cases, the exploration and transformation of conceptual spaces [9]. Either way, such systems have a certain degree of unpredictability for the output to become more exciting or surprising [35]. As for the computational techniques adopted in these systems, there is a wide range of diversity. For example, some of the forms took rule-based systems [34,36], while others, interactive evolutionary systems, such as the one presented by Karl Sims and Moroni [11].

On a multidisciplinary approach, Moroni et al. [3] proposed an interactive evolutionary environment, ArtVox, to evolve images inspired by Kandinsky artworks. This work resulted in the evolution of a visual composition built upon a series of geometric primitives. At the same time, their proposed system aims to contribute to an exploration of the automatic processes of human intelligence and creativity.

Within the field of genetic programming, its methods have been popularly used in creating image generation tools. An example is the NEvAr [37], which proposes a user-guided neuro-evolutionary art system that expresses “the aesthetic and/or artistic principles of the user”. Other approaches to image generation include generative contexts, as illustrated in the field of computer art by the pioneering work of Frieder Nake or Michael Noll [10].

Recent work on evolutionary music by Scirea et al. [38] aimed to create music that could express different moods in a dynamic environment. The system generated compositions with a chord sequence, a melody, and an accompaniment [38], dealing with a set of musical properties, such as harmony, melody, pitch, scale, intensity, timbre, and rhythm.

In sum, for the computational generation of multi-domain associations, we consider the fittest approaches to be driven by rule-based programming, and bio-inspired computation [35,39].

## 4. The Framework

The present framework is a guideline for the computational exploration of cross-modal associations interrelating images, music, and cognitive science findings. A challenge in developing this framework was finding the best combination of theoretical knowledge and computational setups to originate configurations with a certain degree of creativity and resilience [35]. Other challenges included the search for a solution that would allow mappings in the direction of music to image and vice versa. The proposed framework can be consulted in Figure 4. Although in the present framework, we focus on exploring a dialogue between a set of auditory properties and visual properties steered by emotions, we can generalize to other problems for creative purposes. For example, we could explore relationships between a set of human interactions and a set of visual properties as long as a common vocabulary grounded them.

In this framework, the AV language is comprised of a vocabulary and a grammar. The vocabulary has three main alphabets: visual elements, music elements, and basic general emotions. The grammar is responsible for representing associations among elements of our vocabulary set. In this case, MG and IG act as interdependent co-creational processes.Given a pre-defined set of relationships between emotions and music/visual elements, we are able to generate music or image based on a specific context. The AV language can be explored and enhanced through an EE. The evolution of the elements’ combinations will depend on how the system is defined: automatic or interactive. While an automatic evolution is developed to have a system fully autonomous in its decision-making process [40], in future work, the introduction of user input will fall into an interactive evolutionary approach [11].

Ultimately, our aim is that this framework brings new possibilities for expressing AV mappings at a perceptual level. In the following subsections, we further detail each module.

### 4.1. Module I: Vocabulary on the Expressive and Affective Qualities of Image and Music

In this module, we detail the process of finding a common vocabulary to connect auditory and visual domains at the same level. Followed by this, we propose a selection of properties and emotions that are relevant from a perceptual point of view. Figure 5 provides an overview of this process of selection.

To ensure that music and image vocabulary had the same levels of depth, we defined three levels: perceptual category, property, and value. For example, music vocabulary can be unfolded in this way: (1) perceptual category: temporal organization; (2) property: tempo; and (3) values: fast, moderate, and slow. On the other side, the image vocabulary can translate into the following example: (1) perceptual category: low-level features; (2) property: size; and (3) values: big, medium, and small.

We included in the vocabulary both “isolated” and “composite” features, i.e., properties that can be perceived in “isolated” conditions (for example, pitch in music and shape in image) and features, where its EP highly depends on the relationship with or other elements (for example, harmony in music and complexity in image).

The **music vocabulary** was grouped into four perpetually distinct sections: (1) harmony and melody, (2) dynamics, (3) texture, and (4) temporal organization. These parameters were structured according to a perceptual organization suggested by several authors [41,42,43]. Each perceptual category had a set of properties and values associated with it. Below are some auditory samples to exemplify how they can fit into the several music categories. We shall note that they are not presented in the full detail found in the literature, as this is a mere demonstration example.

A.
*Harmony and Melody*
Consonance: https://freesound.org/people/Lemoncreme/sounds/186942, (accessed on 19 September 2022)Dissonance: https://freesound.org/people/ZeritFreeman/sounds/472315, accessed on 19 September 2022)B.
*Dynamics*
Loudness: https://freesound.org/people/NoiseCollector/sounds/15705, (accessed on 19 September 2022)C.
*Texture*
https://freesound.org/people/Angel_Perez_Grandi/sounds/47382, (accessed on 19 September 2022)D.
*Temporal Organization*
Rhythm: https://freesound.org/people/iut_Paris8/sounds/466770, (accessed on 19 September 2022)

As for the **image vocabulary**, it was chosen based on the visual element’s relevance to the creation of expressive visual communication and to the composition of an abstract image. We grouped visual features into four distinct sections: (1) high-level features, (2) low-level features, (3) object manipulations and (4) relationships among elements. Our choice for organizing visual attributes is justified by Christian Leborg’s proposed visual grammar [44]. Additionally, we completed the setlist of attributes with references from literature studies featuring abstract visual alphabets [14,31,45].

In Figure 6, we show a Kandinsky painting to demonstrate how the visual features that we collected can be used to compose an abstract image.

The choice of **emotions vocabulary** relied on finding a set of emotions common to the visual and auditory domains. To assess the maximum of the emotions that are commonly perceived in both visual and auditory domains, we made a list of the emotion-based studies containing them. Several authors often presented similar terminologies during this process, so we opted to merge the synonymous emotion terms. We looked for a dictionary definition to ensure that the words were synonyms. For example, dignity, solemn and serious were all placed in the same category (emotion nº4 of the list enumerated above). In the cases where the same pairs of emotion–feature associations were present in several articles by the same author, we did collect the emotion nomenclature of the most recently published reference. The identified and assembled emotions fall into 16 distinct groups to limit the extensive categorization of emotions. This ended in a set of 16 emotion labels: (1) anger; (2) boredom; (3) calm; (4) dignity; (5) disgust; (6) energy; (7) excitement; (8) expectation; (9) fear; (10) happiness; (11) masterful; (12) pleasantness; (13) sadness; (14) sleepy; (15) surprise; and (16) tenderness.

The resulting AV properties (see Section 2) are categorized into two types: the ones we use to compose music or image in the generator (c), and those we use to explore a set of relationships in the evolutionary engine (e)(see the actions in Figure 5 and data presented Table 6). This separation of properties was necessary to distinguish the needed from the composition’s expressive point of view and the evolutionary engine’s need to adapt and present different representations of emotional states. Consequently, the latter is considered the most perceptually relevant in the AV relationship.

The following table presents an overview of the properties used to compose music and images and those specific to exploring emotions representations in the EE.

As for the process of generating music or images, they are composed of two main steps: definition of grammar rules and interpretation of the rules instantiated. In addition to this, we introduce two concepts: dynamic vectors and static vectors. A dynamic vector is a vector that will change or adapt to a certain context, that is, its values are not fixed (e.g., evolution purpose values). A static vector corresponds, for example, to a set of values that were pre-defined and do not change with different contexts, such as the values used for composition purposes only (see Figure 7). In the following modules, Module II and Module III, we detail the grammar rules defined for representing each domains’ expression and exemplify a possible interpretation of these rules.

### 4.2. Module II: Generation of Music Based on Emotions

The generator of music is an instance of a system previously developed by Seiça et al. [47]. This generator is built as a rule-based system through probabilities and is guided by two significant musical aspects: harmony and melody.

For the properties aimed only at composition (C) purposes, we assigned a neutral state as follows:*Melody intervals* have a 5% for a 1st, 35% for a 2nd, 30% for a 3rd, 5% for a 4th, 10% for 5th, 5% for a 6th, 5% for a 7th, 5% for an 8th.*Melody direction* has 50% probability to be ascending, and 50% to be descending.*Tonality* always follows the western tonal system rules.*Articulation* has 50% probability to be legato, and 50% to be staccato.*Timbre* was simplified, opting for a piano and a violin. The first to play both the melody and the harmony, and the second for the melody. The choice to reduce the number of timbres to just two instruments, whose sound is familiar and well-recognized, was made to balance the tone influence in the emotion association.*Rhythm* in melody was defined by whole notes with 10% of occurrence, half notes with 20%, quarter notes with 50%, and eighth with 20%.*Patterns* occur in melody sequences and harmony progressions within an emotional state.

According to the probabilities defined for each property, the generator shapes the music composition.

The emotion associations with the music properties aimed at the evolutionary exploration of relationships (E) can be consulted in Table 7.

To reduce the complexity of discrete values, we divided them into levels based on the vocabulary defined Module I, as this would provide easier control of the generator. For example, loudness is divided into loud, moderate and soft levels of values. We hereby describe the type of values (levels) that each property (M1, M2, M3, M4, and M5) can take in the MG:*M1 (Pitch)*—low, mid-low, mid-high, and high;*M2 (Sonance)*—consonant, dissonant, and mixed;*M3 (Loudness)*—soft, medium, and loud;*M4 (Tempo)*—slow, medium, and fast;*M5 (Progression)*—angry, calm, happy, and sad pre-defined progressions.

In addition to these probabilities and emotions associations, there were decisions in the generator made in a stochastic way to promote a natural expression (and avoid falling into the rigidity of a ruled-based program). For example, pitch octave, loudness, and tempo values vary slightly within a level. We should note that MG does not have a beginning or an end. This is, it is constantly generating music in real time. For more details on the MG, please consult reference [47].

We provide in the following links to two samples from the MG for the emotions of happiness and sadness produced by our generator:
(a)Happiness: https://www.dropbox.com/s/9d3dvkxy5p5k0mu/happiness.m4a?dl=0, (accessed 19 September 2022)(b)Sadness: https://www.dropbox.com/s/y1ktt8292u00mmt/sadness.m4a?dl=0, (accessed 19 September 2022)


### 4.3. Module III: Generation of Abstract Images Based on Emotions

This module presents a computational algorithm for generating abstract visual compositions.

Regarding the IG, we built a composition with a vocabulary of simple or geometric elements to make it easier to achieve and play with [48]. All images follow a two-dimension static abstract composition, as its subjectivity was suggested by Melcher et al. [5] to have enhanced potential in the representation and perception of distinct emotions.

As we wanted the resulting compositions to be more expressive than the sum of isolated properties, we drew a central concept to developing these images: depth layers (foreground and background). In comparison, the first layer is concerned with directly interpreting isolated properties of any form (low-level features), while the second deals with the relationship between all forms present in a composition (high-level features).

For the properties aimed only at composition (C) purposes, we assigned a neutral state as follows:*Thickness*: it is given a medium-thin stroke value so it does not impact on the emphasis of the visual elements.*Closure*: it is used for foreground and overlapping objects as closed and on background objects as open.*Opacity, Transparency*: Opacity is used in background overlapping objects—they have a 50% chance of being filled with color. As for transparency, it is used in the composition objects to allow a better visibility of the foreground and background layers.*Movement*: although these images are static, the placement of objects in diagonal is desired to achieve the sensation of movement.*Depth*: images feature types (low-level and high-level) are placed in two layers—foreground and background, respectively.*Variation*: Variation is achieved through generativity in some properties. For example, we vary the selection of shapes and the color tones within their levels.

The emotion associations with image properties can be found in Table 8.

**Table 8 entropy-24-01706-t008:** Set of associations between a selected set of visual properties and emotions.

Emotion	V1: Regularity	V2: Edge (fg)	V3: Edge (bg)	V4: Size	V5: Color	V6: Crowdedness	V7: Complexity	V8: Visual Distribution	V9: Orientation
Angry	irregular	sharp	sharp	big	red+black	crowded	complex	*disordered	*diagonal
Calm	regular	round	round	big	green+blue	clear	simple	*structured/disordered	*horizontal
Happy	reg/irrg	round	round	big	yellow+orange	*clear/med	simple	*disordered	*diagonal
Sad	*irregular	round	round/sharp	*small	blue+purple	*crowded	simple/complex	*structured	*vertical
**Neutral**	reg/irrg	round	round	med	grey (light)	clear	simple	struc/disor	horizontal

Note: Associations marked with an asterisk (*) mean that there were not enough literature findings to support this relationship. For this reason, we rely on a previous study [49] and on visual art perception studies [14].

Similar to the music module, to reduce the complexity of discrete values, we transformed it into levels based on the vocabulary of Module I. As a result of this, we describe the type of values (levels) that each visual property (V1, V2, V3, V4, V5) can take in the generator:*V1 (Regularity)*—regular, irregular;*V2 (Foreground Edge)*—round (circle or ellipse), sharp (square or triangle);*V3 (Background Edge)*—round (point or arc), sharp (line or arrow), and *(Background Overlap)*—round (circle, oval or flower), sharp (square, rectangle or star);*V4 (Size)*—small, medium, big;*V5 (Color)*—angry, calm, happy, sad pre-defined colors;*V6 (Crowdedness)*—clear, mid-clear, mid-crowded, crowded;*V7 (Complexity)*—simple, mid-complex, complex;*V8 (Visual Distribution)*—structured, disordered;*V9 (Orientation)*—horizontal, vertical, diagonal;

As the visual interpretation of the properties and values described in here can have some degree of abstractness and aesthetic subjectivity, we exemplify in Figure 8 our visual principles for the composition.

In its current form, the generator of images can be used without a context (e.g., randomly) or given a set of properties and values associated to a specific emotion. For example, provided different combinations of properties and values (see Table 8), for the emotions of happiness and sadness, we are able to generate images as the ones presented in Figure 9:Happiness combinations of values
-**v1:** regular *(set 1)* and irregular *(set 2)*, **v2:** round, **v3:** round(arc), **v4:** big,**v5:** yellow+orange, **v6:** mid crowded, **v7:** simple, **v8:** disordered, **v9:** diagonal.
Sadness combinations of values
-**v1:** irregular, **v2:** round, **v3:** sharp(line) *(set 1, set 3)* and sharp(arrow) *(set 2)*,**v4:** small, **v5:** blue+purple, **v6:** crowded, **v7:** simple *(set 1, set 2)* and mid-complex *(set 3)*, **v8:** structured, **v9:** vertical.


### 4.4. Module IV: An Evolutionary Process for a System of Relationships

The module of EE presents a solution for the last part of the problem: the computational exploration and expansion of the auditory-visual language. We note that a design concern while developing the solution was the possibility of navigating the dialogue in two directions, that is, from music to image or image to music. Given this problem’s abstract and multi-domain nature, we opted for evolutionary programming methods to explore the set of associations.

Generally speaking, evolutionary algorithms work with populations of solutions to help determine where to go next in the space of possibilities. In this process, the computer will be instructed to keep the best solutions and let the worse die. The better solutions are then allowed to reproduce. This simple process causes evolution to occur. A fitness function is typically used to measure the solution’s quality; this is a score of how well the solution fits the objective [50]. Genetic programming, a sub-field of evolutionary algorithms, tries to solve the computational challenge of creating an automatic system whose input is a high-level statement of a problem’s requirements and whose output is a satisfactory solution to the given problem [50]. This seemed to be a good fit for the problem, as it could attend to the design concern of being able to navigate the dialogue in two directions (from music to image or image to music).

For the purpose of this research, our EE was built on top of Tiny GP, an algorithmic implementation of genetic programming proposed by Poli et al. [40]. TinyGP is a simple genetic programming system for symbolic regression. Given a range of input values x, the algorithm’s objective is to find a function f so that f(x) equals a defined target value y. TinyGP finds the function f through evolving equations represented as trees composed of terminals and primitives. This representation is known as a program. The terminal set comprises a range of floating point variables, and the function set comprises addition, multiplication, protected division and subtraction [40].

TinyGP modifies and combines programs, building offspring whose tree terminals and primitives are slightly different from one another and tests each of these programs against the defined target, and therefore evaluating how close each one is to the intended result. The closer the output of a program is to the defined target, the more suitable it is, and therefore, the better its fitness. TinyGP selects programs for modification and combination through tournaments. The algorithm uses point mutation, which means that when a random point is selected, it is replaced by another randomly generated one of the same type. If it is a terminal, another terminal replaces it. If it is a primitive, it is replaced by another primitive. TinyGP uses subtree crossover with a uniform selection of crossover points [40].

Abstractly, TinyGP executes the following steps:Generate an initial population composed of random programsCalculate the output (y) of each program (f(x)) in the population based on a given input (x);Use tournament selection to pick programs;Create a new program through genetic operations and substitute it into the population;While the best fitness is below the threshold of acceptance, go to 2;Return the program with the best fitness.

We expanded Tiny GP to respond appropriately to the needs of our problem. At first, we extended Tiny GP to find a correlation between a vector of N − M variables so that we could characterize music or image expression through a set of parameters (in the original implementation, Poli et al. proposed a correlation of N − 1 variable). This was a critical and necessary change, as it solved the problem of correlating multiple dimensions between the input and the target. Essentially, we create a matrix of two dimensions: one for the number of desired inputs and one for the number of desired output parameters. Then, we have a set of different random variables running through the same program tree for each output.

Secondly, we favored an elitist solution to avoid the loss of the best individual. Third, to avoid premature convergence and a loss of diversity, as we worked with a small population size, we replaced the point mutation (or node replacement mutation) operator with a subtree mutation operator. While in a point mutation, a tree node is randomly selected and then randomly changed (keeping the size of the tree intact), in a subtree mutation operation, a randomly selected subtree of an individual’s program is replaced with another randomly generated subtree.

#### 4.4.1. System Architecture and Performance

We hereby specify the preparatory steps that were applied to our genetic programming algorithm:Set of terminals: set of N music or M image properties and Z random constants.Set of primitive functions: arithmetic functions (+, -, *, /).Fitness measure: distance between the program output and the targeted visual variables.Parameters for controlling the run: population size, subtree crossover probability, subtree mutation probability, program depth, and random seed.Termination criterion and method for designating the result of the run; for example, run 5000 generations.

An overview of the evolutionary engine architecture can be consulted in Figure 10.

Regarding the evolutionary process, an initial population composed of randomly generated programs is created and evolved by executing the following steps until the termination criteria are reached:Calculation of the fitness of each program using a multidimensional approach.Manipulation of programs is based on predefined set probabilities and can fall accordingly into one or more of the three operations:Selection of two programs to perform a subtree crossover.Selection of one program to perform a subtree mutation.Copy of selected programs, making sure that the best individual of the population is guaranteed to be selected.The resulting programs of these manipulations will replace others that were selected by means of a negative tournament, i.e., those with the worst fitness in the population.

Once the termination criteria are reached, the individual with the best fitness is assessed, and its corresponding program is selected as the result of the run. A value of zero indicates that a solution was found.

We ran some tests with several configurations and random seeds to test the ability of our system to evolve. For each input(s), a corresponding set of outputs(s) is evolved (as illustrated in Figure 10) toward a target. The target in this case, is based on theoretical findings (see Table 8). If the system was guided by user interaction, the target to be considered would be considered the user input.

The system is run in automatic mode (see Section 4.4.2). Considering that each individual has four visual expressions, the visual expressions are then composed of a set of properties with corresponding values. With this, we exemplify the calculation of the *fitness* value regarding the translation example from music to image.

The first step is to calculate the distance of current visual properties values to the target values. Here, if the image output value corresponds to the theory value, it is assigned a value of 0 (distP1 = 0). If this is not true, then distP1 = distance between target (theory) and output (program result).This process is repeated for each image property value.We then sum the absolute fitness value for all properties of an image. Then, we perform a division of the previous sum by the number of image properties in such a way that imageFitness1 = distP1+distP2+..distPN/NimageProperties)The fitness value of an individual or program is a division of the previous sum by the number of image expressions: programFitness = (imageFitness1 + imageFitness2 + imageFitness3 + imageFitness4)/4

The following figures (Figure 11 and Figure 12) present the progression of the system performance by presenting a set of graphics the best individuals values in the present conditions:depth: 5positiveTournament size: 5negativeTournament size: 10crossover probability: 0.5mutation probability: 0.6population size: 30number of music properties: 5number of image properties: 9

**Figure 11 entropy-24-01706-f011:**
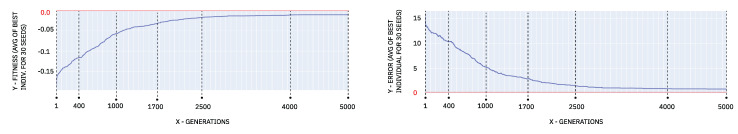
Direction of music − image. The graphic on the left presents a correlation between fitness values and program generations. The graphic on the right presents the number of errors, i.e., number of properties wrong at each generation. In both graphics, the calculations present an average of the fitness of the best individual for 30 seeds.

**Figure 12 entropy-24-01706-f012:**
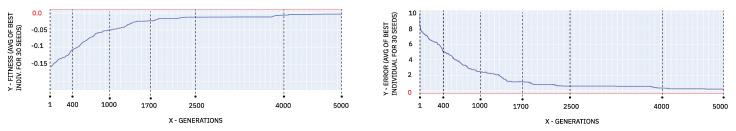
Direction of image − music. The graphic on the left presents a correlation between fitness values and program generations. The graphic on the right presents the number of errors, i.e., number of properties wrong at each generation. In both graphics, calculations present an average of the fitness of the best individual for 30 seeds.

Figure 11 presents the evolutionary system performance when programmed to evolve a set of images given a set of music inputs, that is, in the direction of **music to image**. The error presented in this image corresponds to the sum of the number of wrong properties in the four evolved expressions.

Figure 12 presents the evolutionary system performance when programmed to evolve a set of music given a set of image inputs, this is, in the direction of **image to music**. These programs could find solutions for the problem before the termination criteria were reached, demonstrating that the proposed approach can evolve toward the intended target under different conditions, and that this ability is not limited to a specific set of parameters.

Looking at these figures, it is clear that the fitness function is non-decreasing, i.e., it either remains constant or increases, proving its ability to evolve toward the intended target. In sum, the evolutionary capabilities of the proposed approach favor results in the sense that an extended series of runs were executed successfully, with solutions being found for several instances of an individual. This proves the system’s capacity to adapt an individual’s expression to different emotion contexts.

#### 4.4.2. An Automatic Guided Evolution

We designed a test to evaluate whether our system had the needed requirements to solve the proposed problem. We followed by evolving a translation of music to image. In detail, a set of musical properties was given as input to the system, and then a set of visual properties evolved according to a theoretical target. We measure the success of our approach by proving that our system can perform a translation between distinct domains while dynamically adapting to different testing conditions. That is, the programs in the population can maximize their fitness and adapt to four different types of music to an outcome of four corresponding images with the same emotional expression as the music inputs.

Looking at Figure 13, we can visualize some system outcomes through different generations. Each individual or program is composed of four expressions, presented vertically (A, C, H, and S). The progress of individuals is contrasting when we look at the first and last generations. As the program evolves, individuals start to converge into the targeted solution; therefore, more similar expressions begin to appear. By the last generation (gen 1001), practically all the individuals of the population were able to find the optimal solution. Although we ran the evolutionary with 30 individuals, we only present a smaller subset for example purposes.

The obtained results suggest the tool’s plasticity and capability to express a cross-modal audio–visual language, which can be used in creative contexts. We should emphasize that this implementation goes beyond the original implementation of Poli, hence its novelty. In future iterations of this work, we intend to explore the system with an interactive user-guided approach.

## 5. Experimental Results: An Adaptive Exploration of Auditory–Visual Relationships

### 5.1. Sample Characterization

Seventeen preliminary user tests were performed to evaluate the system’s expressive capacity for the translation between domains. The age of participants ranged from 26 to 35 years old, with a median of 30 years old. Eleven subjects had a background in media design, four subjects had a background in computer science, one had a background in human resources and the other one in healthcare. Although the tests were performed online, the same exact instructions were given to all the participants, and they had remote control of the screen to perform the test.

### 5.2. Evaluation Process and Methodology

To perform the user tests, we conceived an interface allowing natural interaction with the system. The test would begin by presenting a set of four ordered music and a set of images displayed in random order. The four pieces of music presented in the test were previously given as input to our system, and each one had an underlying emotion: angry, calm, happy, and sad. The images presented resulted from the system evolution given these musical inputs. A strategy to try to control the interdependence on perceived emotion and bias relied on randomizing the presented images. Additionally, experiments had a pre-study tutorial phase to remove initial learning.

During the test, participants could listen to every music type as often as needed—they would choose when to play it or not, but we suggested that they begin the test by listening to them all to have an overview. Although music was played generatively, we fixed some parameters to a neutral state. We adjusted the musical parameters that were considered more perceptually relevant in the expression of emotions, such as pitch, sonance, loudness, tempo and progression. The images had a 2D static representation, and all the participants had access to the same evolved images.

Then, the participants were asked to match the pairs of music and image, considering that they should represent or express the same underlying emotion or sensation. Once a participant reached the desired combination of music and image, these were paired side to side. Afterward, we asked participants whether they recognized the underlying emotions and sensations behind the pairs they had connected and asked them to describe each pair.

The music set used on these tests is presented in the following URLs (last access on November 2022):M1—Angry: https://www.dropbox.com/s/e5t57n1004bypjm/anger.m4a?dl=0M2—Calm: https://www.dropbox.com/s/uf4gm2s5u0lhrow/calm.m4a?dl=0M3—Happy: https://www.dropbox.com/s/9d3dvkxy5p5k0mu/happiness.m4a?dl=0M4—Sad: https://www.dropbox.com/s/y1ktt8292u00mmt/sadness.m4a?dl=0

The evolved image set from the input musics can be consulted in Figure 14.

### 5.3. Analysis of Results

We divided the user tests into quantitative and qualitative phases to analyze the results. In the first phase, we present the data resulting from the users’ perception and compare them to the expected mappings of the previous system evolution. In the second part, we analyze the results from a qualitative perspective, including the open answers of the users.

#### 5.3.1. Quantitative Analysis

In Table 9, we show the user tests’ results by comparing the system’s expected mappings with the users’ perception. We use the label ”M” for music and ”I” for image. The numbers 1 to 4 represent each of the emotions: (1) angry, (2) calm, (3) happy and (4) sad. The expected mappings were M1-I1 for angry, M2-I2 for calm, M3-I3 for happy and M4-I4 for sad.

Considering the obtained values, we can understand that the users most easily perceived the outputs did regard the emotions of happiness and anger. It is also noteworthy to point out that 23.5% of participants preferred sad expression to represent calm and 29.4% preferred the calm expression to represent sad. These results can be explained by the energy proximity between the two emotions. Both can be classified as low arousal in the Russell model of emotions [12]. Regarding the less significant result, some users related happy music to the calm image and vice versa.

Since the representation of a set of properties is perceived differently than its isolated expressions, the divergence in participants’ opinions could indicate a need for a more expressive visual representation, or indicate that their visual expression differs from the ones suggested by our dataset. Nonetheless, when comparing the users’ perceptions with the expected results, we can conclude that participants’ agree in the majority of pairs with the system’s evolved outcomes, strongly suggesting the feasibility of our hypothesis. There is, however, space for improvements in the computational expression of these associations.

#### 5.3.2. Qualitative Analysis

The user’s open answers integrated the last part of the test. They suggested that emotions with contrasting valence and arousal—such as happiness and anger—were more easily recognized than the less contrasting ones (calm and sadness). These answers are also suggestive of the following: (i) The mappings of happiness and anger were very clear, being the last one the most clear. (ii) The low complexity of the visual output for calm had a high positive impact on its perception. (iii) Calm and sadness are harder to perceive. (iv) It was easier to acknowledge the underlying emotion through the image than through the music.

Moreover, during the tests, users expressed other feelings and words of EP in the following music (M)–image (I) pairs:Angry (M1–I1): heavy, negative, intense, strong, exaltation, energic, tense, quick, fear, anticipation, aggressive, chaos, and fury.Calm (M2–I2): positive, pacific, peace, joy, introspection, relaxing, expanding, hope, sad, nostalgia, meditation, trust, positive, neutral, elegant, melancholy, sadness, and serene.Happy (M3–I3): happy, joy, positive, anticipation, and hope.Sad (M4–I4): melancholy, negative, subtle, heavy, slow, calm, empty, fear, anticipation, content, depressed, lonely, insecure, lost, and calm.

The qualitative analysis clarified the results obtained in the quantitative analysis in the sense that participants were able to identify the four emotions driving each music–image pair of associations.

Furthermore, users noted a few music and visual properties that drove their choices while assembling music–image pairs. The properties that suggested to have more impact on the participants’ perception when making decisions were loudness, tempo, shape/edge+regularity, color and complexity.

## 6. Discussion and Conclusions

The present research emerged from a need to map distinct creative domains through a common language. Although several examples of AV mappings can be found in the literature, they are often produced non-empirically or with rigid approaches. Given the complexity and abstraction of creative domains, a solution with more space for integrating knowledge from other domains and flexibility is essential for dynamically studying audio–visual languages. For this reason, we propose a multidisciplinary approach to this problem, followed by a methodological process and experimentation. During this process, we address challenges, such as the computational representation of multidimensional data and the dynamics of nature-inspired computation.

Supported by the knowledge that a significant bridge between music and images can enhance our ability to communicate emotions or senses, the study of cross-modal associations between sound and image motivated by EP provides a solid basis for interrelating these domains. Overall, emotions have been well-accepted to represent abstract domains, such as music and image. This way, we could systematize the theoretical findings and build adaptive relationships between the two domains.

The presented solution and framework brought flexibility and dynamics in exploring auditory–visual relationships, as opposed to the works described in the literature, where authors usually rely on a pre-defined static set of relationships. It allowed us to include theoretical findings of perception and emotion and define a shared vocabulary between the two domains. Moreover, the possibility of driving the computational generation of music or image steered by an emotional context is relevant, since research on integrating cognitive science findings with evolutionary strategies for creative purposes is a topic yet to be studied in depth.

From the theoretical analysis of the literature, we can conclude that “sadness”, “happiness”, “fear” and “anger” are the emotions most commonly associated with visual and musical properties, followed by “tenderness” and “calm”. Furthermore, we can conclude that the most perceptually relevant visual properties to express emotions are color (brightness, hue, and saturation), shape, edge, size and texture, and that the most perceptually relevant musical properties to express emotions are tempo, loudness, articulation, high-frequency energy and harmony sonance.

The performed user tests favored the integration of emotion and perception findings in the representation and abstraction of auditory and visual languages, as the evolved expressions were compatible with users’ perception of emotions. For instance, angry and happiness were the participants’ most easily perceived music–image pairs, while calm and sadness had more divergent perceptions. User studies regarding music and image are a challenging and delicate topic experimentally, as they suffer from two core complications. First, we would need an independent measure of what emotions are. Furthermore, it is known that subjects disagree about perceived emotions; hence, we cannot measure ground truths, only provide tendencies. Secondly, music and image complexity offers a rich array of cross-effects and biasing.

Nevertheless, we offer a systematic approach to a multidisciplinary problem of creative nature and contribute to a better understanding of the transposition of human creative processes in computational approaches. In future, this work can be improved by performing experiments with a broader set of music and image properties regarding the EE module and experimenting with other computational techniques to generate abstract images, such as shape grammars [51]. Although some recent approaches have used neural networks to generate images or music, they remain a black box concerning decision making.

Relying on explainable AI techniques, a recent field dedicated to investigating the design of computational systems so that the processes are comprehensible to both humans and machines [52], we propose to study new instances of our solution further. To achieve this, we will build a partially guided artifact, where the user can guide the system and customize the audio–visual associations. Furthermore, this user study shall be carefully designed to evaluate the system’s capacity to adapt to user input, provide personalized image–music pairs based on the user’s perception of audio–visual representations, and unveil tendencies that most impact this exploration.

On an end note, this article provides key concepts and findings with the potential of application to several groups of interest, aiming to contribute to solve problems in visual communication, music communication, audio–visual artifacts, and behavioral sciences (mood control and group identity). However, detailing the application of our research to the fields mentioned above is out of this article’s scope, as it would require a further study of their problems and state-of-the-art re-contextualization.

## Figures and Tables

**Figure 1 entropy-24-01706-f001:**
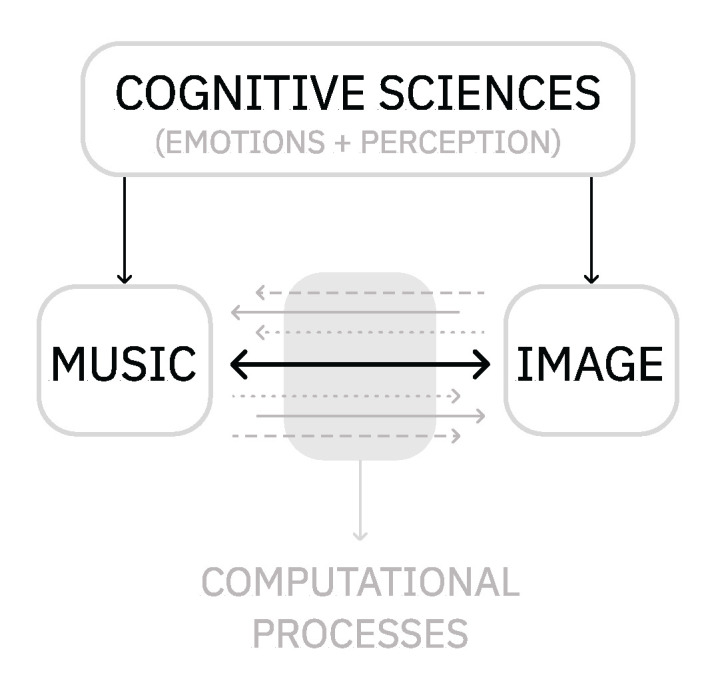
This figure is a generic illustration of the hypothesis addressed in this article. We propose studying a set of relationships that may emerge from crossing music, image, and cognitive sciences, aided by the capacity of computational processes to abstract this problem.

**Figure 3 entropy-24-01706-f003:**
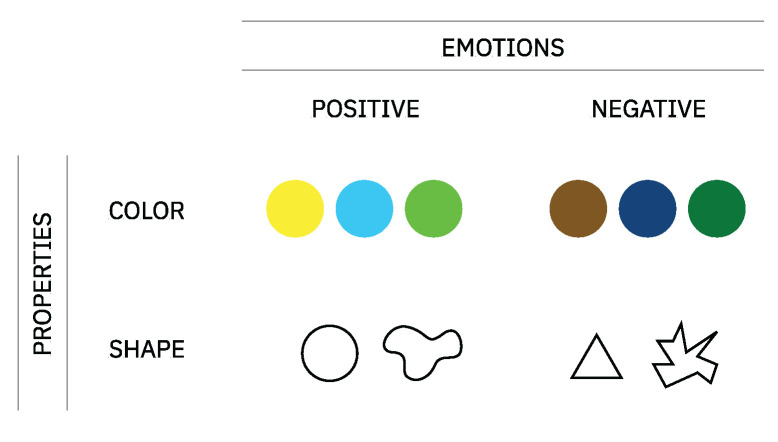
Examples of associations between visual characteristics and emotions.

**Figure 4 entropy-24-01706-f004:**
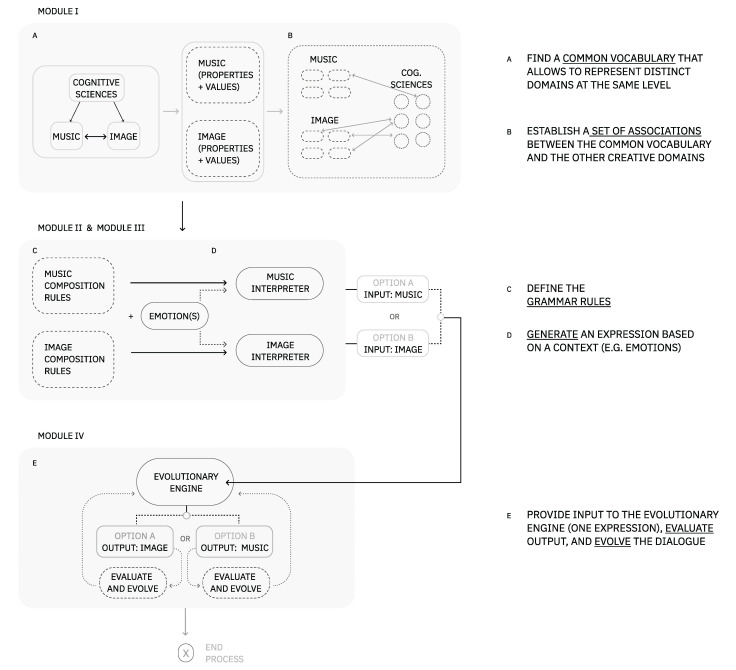
The present framework comprises four modules: a vocabulary, an MG, an IG (image generator), and an evolutionary engine. In this figure, we can observe a series of steps connecting the framework modules that, in turn, work as a guide to this process of development.improved figure.

**Figure 5 entropy-24-01706-f005:**
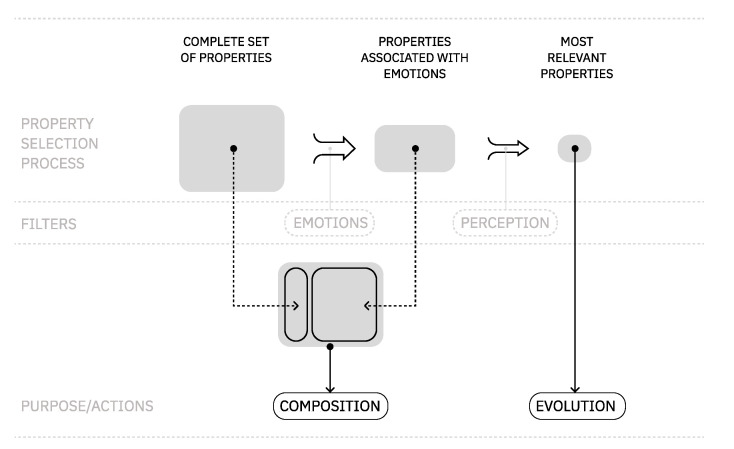
Overview of the properties selection process. This process was applied to both visual and auditory domains.

**Figure 6 entropy-24-01706-f006:**
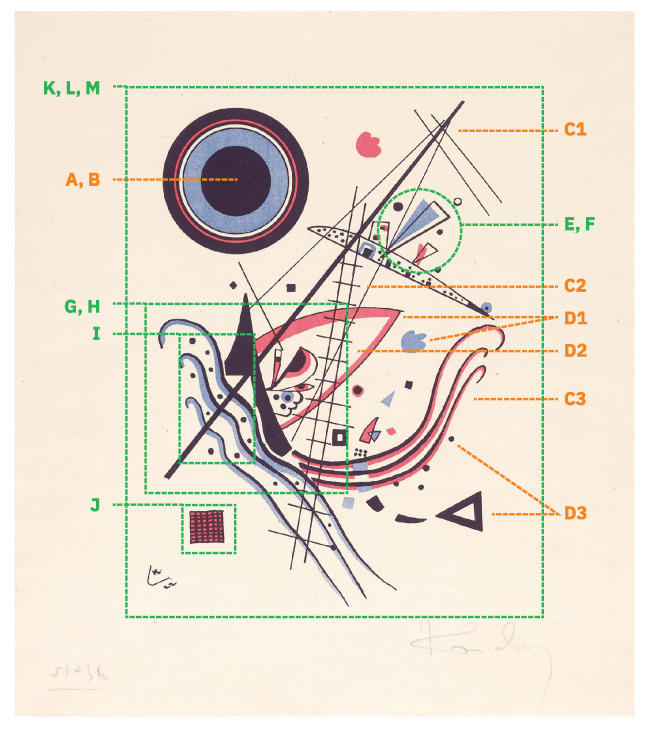
Visual analysis of Kandinsky painting, “Blue”, 1922. The orange lines concern “low-level” or “isolated” features and the green lines concern “high-level” or “composite” features. The letters in the image exemplify the following properties: (A, B) size and color; (C1, C2, C3) line thickness, regularity, and curve; (D1, D2, D3) shape contours, closure, and type; (E, F) elements overlap and geometric manipulations; (G, H) density and complexity of elements; (I) repetition of elements; (J) texture; (K, L, M) visual distribution, composition crowdedness, and variety of elements.

**Figure 7 entropy-24-01706-f007:**
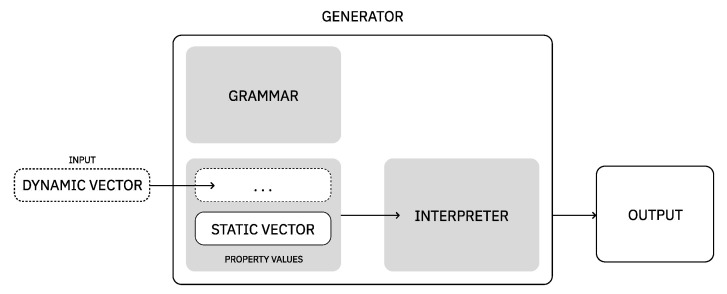
Overview on the process of generating a music or image. In order to generate one composition (output), a vector of values (input) needs to be provided.

**Figure 8 entropy-24-01706-f008:**
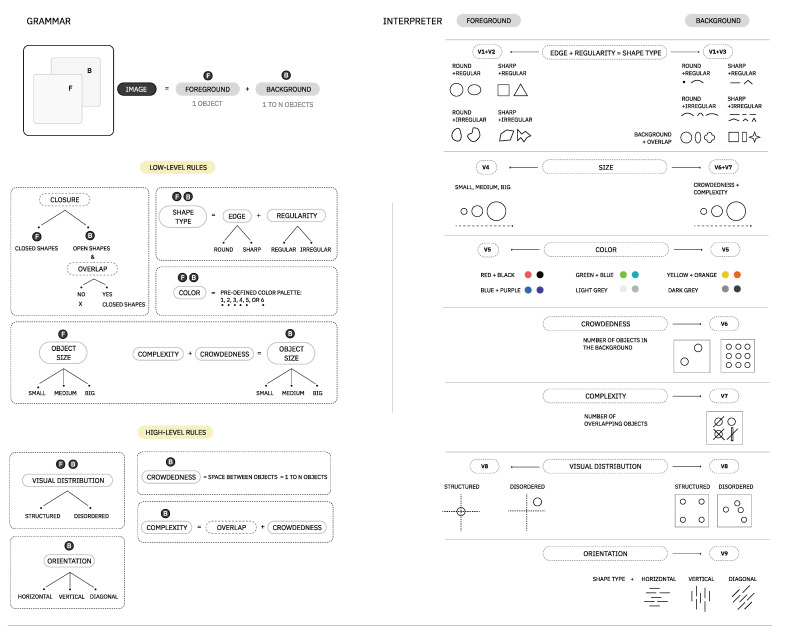
This figure is a proposal by the authors of a set of rules and expression interpretation to generate images dynamically, given a set of properties and values (example in Figure 9). All images are composed of two layers: foreground and background. (**Left**) Visual grammar rules examples for each layer of the image. (**Right**) Illustrated interpretation of grammar rules’ instances.

**Figure 9 entropy-24-01706-f009:**
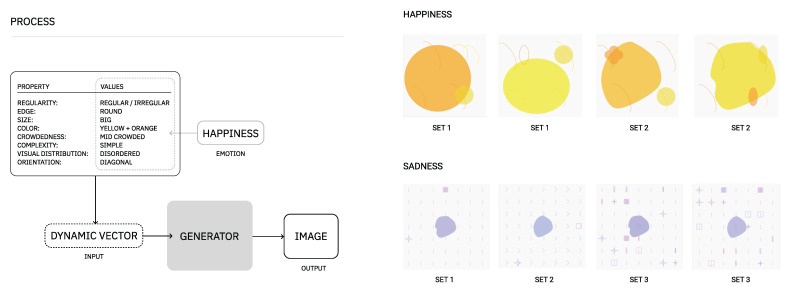
Examples on the process (**left**) of generating visual representation of happiness and sadness emotions (**right**). As it can be noticed, the image generator has a certain degree of generativity; it should not compromise, however, the perception of the emotion.

**Figure 10 entropy-24-01706-f010:**
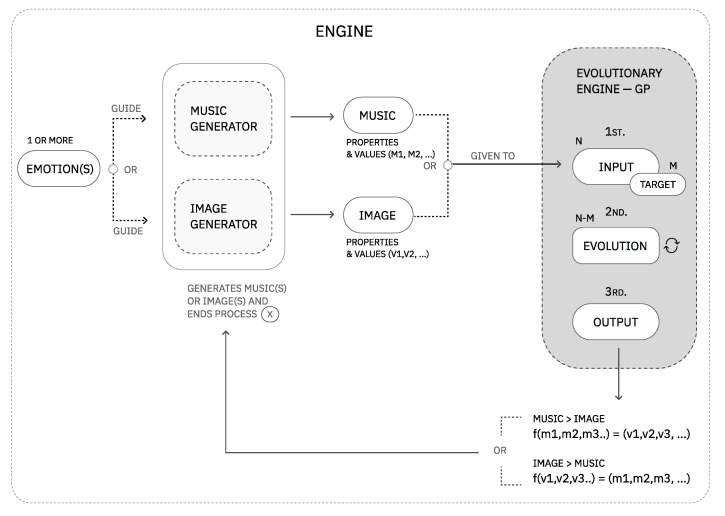
Overview of the EE architecture. A and B phases can be found in more detail in Module I, Module II and Module III of the framework.

**Figure 13 entropy-24-01706-f013:**
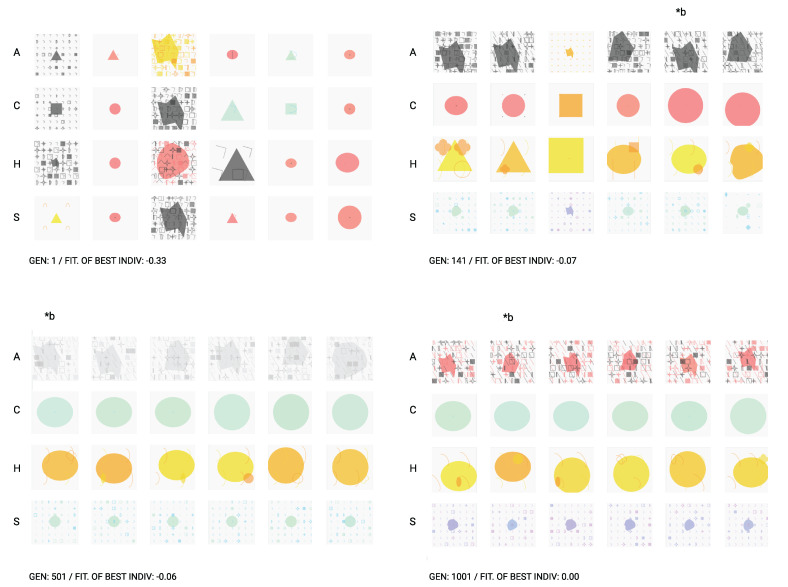
Samples of individuals at several generations. Each individual has four emotion expressions—represented in columns. The letters A, C, H, and S (angry, calm, happy, and sad) stand for the emotions of each row. The designation *b stand for the representations of the best individual. At generation 1001, the algorithm finds the perfect solution (fitness = 0.00) for the problem described.

**Figure 14 entropy-24-01706-f014:**
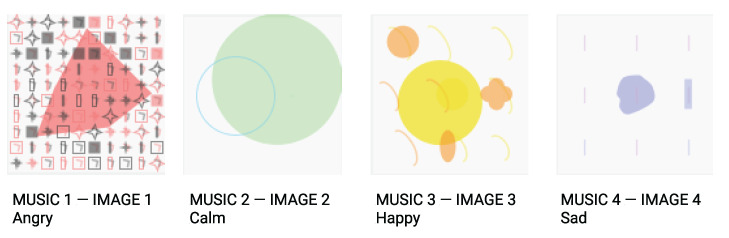
Expressions of the four evolved images to each music–emotion input.

**Table 1 entropy-24-01706-t001:** The table on the left (Auditory Elements) presents the relationships found in literature between auditory elements and emotions. The table on the right (Visual Elements) presents the relationships found in the literature between visual elements and emotions. We restrain this table data to a set of four emotions. Every relationship mentioned in literature is marked with a cross (x). Whenever there are concurrent findings, that is, two or more property values are assigned to the same emotion, we mark with an asterisk (*) the relationship found to have more empirical support. In the remaining concurrent cases, the ones with equal empirical weight, we restrain to the cross.

Auditory Elements			Emotions				Visual Elements			Emotions			
Perceptual Category	Property	Values	Angry	Calm	Happy	Sad	Perceptual Category	Property	Values	Angry	Calm	Happy	Sad
Harmony and Melody	Harmony Sonance	consonant		x	x		Low-Level features (Elements, Objects)	Shape	line				x
		dissonant	x			x			point				
	Melodic Intervals	m 2nd				x			circle		x	* x	
		m 7th			x				oval		x		x
		M 3rd			x				triangle	x		x	
		M 6th			x				square	x			
		P 4th			x				rectangle				* x
		P 5th			x				pentagon	x			x
		Ag 4th	x						star			x	
	Melodic Direction	ascending	x	x	* x	x	Low-Level features (Elements’ Characteristics)	Edge	sharp, angular	* x			x
		descending			x	x			round, curve	x	x	x	*x
	Pitch Level	high	x	x	x			Curve	wavy	x			
		low		x		x			straight	x	x	x	
	Pitch Variation	large	x		x			Tickness	thin			x	
		medium	x						medium	x		x	x
		small	x		x	x			thick		x		x
	Pitch Range	wide			x			Size	big	x	x	x	
		narrow		x		x			medium				x
	Mode	major		x	x				small				* x
		minor	x			x		Color (hue)	red	* x		* x	x
	Tonality	atonal	x						orange	x	x	x	
		tonal		x	x				yellow	x	x	* x	x
Dynamics	Amplitude Envelope	round				x			green		* x	x	x
		sharp	x		x				blue		* x	x	* x
	Articulation	staccato	* x						violet				x
		legato	x	x	x	x			purple		x		x
	Articulation Variation	large			x				pink		x	x	
		medium	x						brown	x	x		x
		small				x			grey/dull		x		x
	Loudness	loud	x		* x				black	x			* x
		moderate			* x	x			white		x	x	x
		soft			x	* x		Color (brightness)	bright	x	x	x	x
	Loudness variattion	large	x		* x	x			dark	* x			* x
		moderate			x	x		Color (saturation)	vivid	x		x	x
		small	x		* x	* x			low	x			x
Texture	Timbre	many harmonics	x		* x		High-Level features	Complexity	simple			x	x
		few harmonics			x	x			complex	x			x
	High-frequency energy	high	x		x			Crowdedness	clear		x	x	
		medium			* x				crowded	x			
		low			x	x		Texture	soft		x	x	x
	Tone-attack	fast	x		* x				rough	x			* x
		slow			x	x	Manipulations	Movement Direction	up	x		x	
Temporal Organization	Rhythm	regular, smooth		x	x				down		x		x
		irregular, rough					Relations	Weight	light		x	x	x
		complex	x						heavy				x
		varied			x								
		firm				x							
		flowing		x	x								
	Tempo	fast	* x		* x								
		moderate	x		x								
		slow		x		x							
	Tempo variation	large				* x							
		medium	* x		x	* x							
		small	x		* x	x							
	Contrast in note duration	sharp, big	x		x								
		soft, small			x	x							

Note: The material used to draw the relationship between music elements and emotions can be consulted in these references [23,29,30]. The material used to draw the relationship between visual elements and emotions can be consulted in references [14,25,27,31,32].

**Table 2 entropy-24-01706-t002:** Number of times a musical property was associated with the emotion “Anger”.

Emotion: Anger								
Harmony sonance	Harmonic intervals	Melodic intervals	Melodic pitch range	Pitch level (height)	Melodic direction	Melodic motion	Pitch variation	Mode
3	0	1	0	1	2	0	2	1
Tonality	Amplitude envelope	Articulation	Articulation variation	Loudness	Loudness variation	Vibrato	Timbre	High-freq. energy
1	3	15	6	25	6	8	10	10
Tone attack	Pause	Rhythm	Tempo	Tempo variation	Note density	Pattern	Contrast (long and short notes)	
9	0	2	25	5	0	0	5	
Total Associations: 140								

**Table 3 entropy-24-01706-t003:** (Left) Total sum of emotions associations considered in the study of visual properties. (Right) Total sum of emotions associations considered in the study of visual properties.

Auditory Emotions	Sum of Associations (Emotion-Properties)	Visual Emotions	Sum of Associations (Emotion-Properties)
Anger	140	Anger	39
Calm	17	Boredom	14
Happiness	192	Calm	34
Sadness	175	Dignity	12
Boredom	8	Disgust	4
Dignity	36	Energy	21
Disgust	9	Excitement	28
Energy	37	Expectation	1
Excitement	9	Fear	41
Expectation	3	Happiness	63
Fear	95	Masterful	14
Masterful	23	Pleasantness	34
Pleasantness	25	Sadness	68
Sleepy	5	Sleepy	13
Surprise	9	Surprise	2
Tenderness	73	Tenderness	22

**Table 4 entropy-24-01706-t004:** Amount of association between visual property “color” and several emotions.

Color: Brightness, Hue, Saturation							
Anger	Boredom	Calm	Dignity	Disgust	Energy	Excitement	Expectation
20	11	23	12	4	12	17	0
Fear	Happiness	Masterful	Pleasantness	Sadness	Sleepy	Surprise	Tenderness
30	36	14	12	36	4	2	19
Total Associations: 252							

**Table 5 entropy-24-01706-t005:** Most relevant properties associated to emotions. (**Left**) Auditory properties and (**Right**) visual properties.We highlight in gray the ten most relevant properties for the representation of emotions.

Auditory Properties	% Property Relevance	Visual Properties	% Property Relevance
Tempo	21. 26	Color (h.s.b)	61.47
Loudness	14.37	Shape	6.83
Articulation	5.49	Edge	5.37
High-frequency energy	4.91	Size	3.42
Harmony Sonance	4.56	Texture	3.42
Pitch level (height)	4.55	Thickness	3.17
Loudness variation	4.44	Movement direction	2.93
Mode	4.32	Curve	1.95
Tempo variation	3.86	Weight	1.95
Timbre	3.51	Closure	1.47
Tone attack	3.39	Complexity	1.22
Vibrato	3.38	Crowdness	1.22
Melodic direction/Pitch contour	3.04	Density	0.98
Amplitude envelope	2.81	Movement	0.98
Articulation variation	2.8	Augemnt/Diminish	0.97
Contrast between long and short notes	2.46	Regularity	0.97
Rhythm	2.45	Position. Orientation	0.49
Pitch variation	2.34	Repetition	0.49
Melodic intervals	2.33	Volume	0.48
Melodic (pitch) range	1.52	Symmetry	0.24
Harmonic intervals	0.7	Cluster. Group	0
Tonality	0.59	Balance	0
Melodic motion	0.47	Depth	0
Note density	0.35	Distance	0
Pattern	0.12	Opacity. Transparency	0
Pause/Rest	0	Overlap	0
		Rotation	0
		Variation. Diversity	0
		Visual distribution	0

**Table 6 entropy-24-01706-t006:** Table of auditory and visual properties selection, type of use (composition or evolutionary), and an interpretation or perceptual effect on the use of these properties. We later use the perceptual interpretation as a guideline in the generation of music and image expressions. Moreover, we identify the properties’ (*M’s and *V’s) that are mentioned in the following Table 7 and Table 8.

Perceptual **Category**	Auditory Properties	Use Type (C: Composition, E: Evolution)	Perceptual Interpretation	Perceptual Category	Visual Properties	Use Type (C: Composition, E: Evolution)	Perceptual **Interpretation**
Harmony and Melody	Harmony Sonance **M2*	E	consonant → dissonant, simple → complex	Low-Level features (Elements)	Shape **V1 + (V2, V3)*	E	geometric → organic
	Harmonic Intervals **M5*	E	positive → negative	Low-Level features (Elements’ Characteristics)	Edge **V2, *V3*	E	round → sharp
	Melodic Intervals	C	order → chaos		Regularity **V1*	E	regular → irregular, soft → rough
	Melodic Direction	C	ascending → descending		Thickness	C	thin → thick
	Pitch Level **M1*	E	low → high		Closure	C	open → close
	Mode **M5*	E	positive → negative		Size **V4*	E	small → big
	Tonality	C	tonal → atonal		Color (hue, sat., brigh.) **V5*	E	colored → monochrome, vivid → pale, bright → dark
Dynamics	Articulation	C	legato → staccato		Opacity	C	transparent → opaque
	Loudness **M3*	E	soft → loud	High-Level features	Complexity **V7*	E	simple → complex
Texture	Timbre	C	soft → rough		Crowdedness **V6*	E	clear → crowded
Temporal Organization	Rhythm	C	note duration, sparse → dense		Orientation **V9*	E	stagnation → dynamic, direction
	Tempo **M4*	E	slow → fast		Visual Distribution **V8*	E	balance → tension, structured → disordered
	Pattern	C	sequence → repetition	Manipulations	Rotation **V9*	E	orientation
					Movement	C	static → motion, slow → fast
				Relations	Depth	C	background → foreground
					Variation	C	homogeneous → heterogeneous, repetition → contrast
					Overlap **V7*	E	simple → complex

Note: The research references used to extract the perceptual interpretation of these properties can be found in references [6,14,25,31,32,46].

**Table 7 entropy-24-01706-t007:** Set of associations between a selected set of music properties and emotions. Although neutral is not an emotional state that we collected from literature, we define it for user test purposes.

Emotion	M1: Pitch	M2: Sonance	M3: Loudness	M4: Tempo	M5: Progression (Intervals + Mode)
Angry	low	dissonant	loud	fast	|| I7 | bII o | I7 | bII o | | II7 II7 bIII o III 7ALT ||
Calm	mid-high	consonant	soft	slow	|| III-7 VI7 | II-7 V7 ||
Happy	high	consonant	mid-loud	fast	|| II-7 | IV-7 | I ||
Sad	low	dissonant	soft	slow	|| I-7 | IV-7| I-7| V-7 IV-7|I-7||
**Neutral**	med	consonant	med	med	|| III-7 VI7 | II-7 V7 ||

**Table 9 entropy-24-01706-t009:** Quantitative results from the user tests. (N) represents the number of participants. Then we present the pairs of music (M’) and image (I’) evolved. In the remaining table, we present data regarding the standard deviation, the percentage of answers for each mapping that was in agreement (E = U), each one that was in disagreement (E!=U), the suggested mapping when E! = U, and the percentage of situations when users did not find a corresponding mapping.

N Participants	Emotion	Music M’ for Each Emotion	Expected Image I’ for Each M’	Standard Deviation	Expected = User (%)	Expected != User (%)	Expected != User (suggested I’ for M’)	No Pair Found (%)
17	Angry	M1	I1	0	100	0.0	n.a.	0.0
17	Calm	M2	I2	1.13	58.8	41.2	I3 (11.8%), I4 (23.5%)	5.9
17	Happy	M3	I3	0.9	82.3	17.7	I2 (5.9%), I4 (5.9%)	5.9
17	Sad	M4	I4	0.79	64.7	35.3	I2 (29.4%), I3 (5.9%)	0.0

## Data Availability

Not applicable.

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
