# Peer review of "“Found in Translation”: An Evolutionary Framework for Auditory–Visual Relationships"

_entropy, 2022, doi:10.3390/e24121706_

Round 1
Reviewer 1 Report
This paper may provide key concepts and findings to several groups of interest aiming to solve problems in visual communication, music communication, audio-visual artefacts, and behavioral sciences (mood control, group identity).
In future iterations of this work, the reseachers can give the user the ability to guide the system and customize the audio-visual associations. Also, a study should be performed to analyze the associations that may emerge from a user-guided system evolution and compare them to some theoretical findings on this topic.
Author Response
- "This paper may provide key concepts and findings to several groups of interest aiming to solve problems in visual communication, music communication, audio-visual artefacts, and behavioral sciences (mood control, group identity)."
- Response 1: Detailing the application of our research to the fields mentioned above is out of this article’s scope as it would require a further study of their problems and state of the art re-contextualization. At the end of page 25 of the article, a more detailed note justifying this response can be found.
- "In future iterations of this work, the reseachers can give the user the ability to guide the system and customize the audio-visual associations. Also, a study should be performed to analyze the associations that may emerge from a user-guided system evolution and compare them to some theoretical findings on this topic."
- Response 2: It is unclear what we could improve in the article give the reviewer phrase, as we had previously stated this in the article. We provided however a little more detail on future iterations of this work in the discussion section.
- Response 2: It is unclear what we could improve in the article give the reviewer phrase, as we had previously stated this in the article. We provided however a little more detail on future iterations of this work in the discussion section.
English language and style corrections have been addressed by revising the written text in the document.
* All the revisions performed to the article have been highlighted in green so that reviewers can easily track the changes.
Reviewer 2 Report
October 31, 2022
Review of the manuscript entitled:
“Found in Translation”: An Evolutionary Framework for Auditory-Visual Relationships
By Ana Rodrigues, Bruna Sousa, Amílcar Cardoso, Penousal Machado
Manuscript ID: entropy-1954637
General Comments
This article presents a new approach to the computational treatment of the relationship between music and abstract images based on the perception of emotions. To solve this complex problem a new idea for the development and study of cross-modal languages ​​ between music and image is proposed. It consists of a set of stages with four main modules: Vocabulary Pack, Music Generator, Image Generator and an Evolution Algorithm.
It is a very interesting and important problem in a multidisciplinary sense, and Neuroscience and Psychology are among them.
Major Comments
It's a very interesting article and maybe even an inspirational article, but I have the following comments and concerns:
1. The main conclusions should be given in explicit form (maybe even in point by point form). These conclusions should be also announced in the Abstract.
2. Some graphic illustrations (diagrams, graphs etc.) should be useful to clarify/explain basic idea and results of the paper.
3. The idea of application of Evolutionary Algorithm should be explained in a more detailed way (application of mutation operator etc.). Some brief recalling of the Tiny GP algorithm should be given. More clear explanation (including a graphical illustration) of the adaptation of this algorithm to the problems under consideration should make the paper more practical.
Minor Comments
a) English should certainly be improved.
b) Editorial form of the paper must be improved.
Final Comments
The article deals with a very interesting problem, and the developed idea is very promising. However, in view of the above comments, in my opinion, this publication may be considered for publication, provided that the above concerns will be carefully addressed. I recommend possibly a Major Revision.
Author Response
- "The main conclusions should be given in explicit form (maybe even in point by point form). These conclusions should be also announced in the Abstract."
- Response:conclusions were made more explicit both in the abstract (see page 1) and discussion section (see pages 24 and 25)
- Response:conclusions were made more explicit both in the abstract (see page 1) and discussion section (see pages 24 and 25)
- "Some graphic illustrations (diagrams, graphs etc.) should be useful to clarify/explain basic idea and results of the paper."
- Response: we added the following figures to the paper trying to clarify the concepts involved and some processes: Figure 1, Figure 2, Figure 3, Figure 5, Figure 7, Figure 8, Figure 9. We improved Figures 4 and 10.
- Response: we added the following figures to the paper trying to clarify the concepts involved and some processes: Figure 1, Figure 2, Figure 3, Figure 5, Figure 7, Figure 8, Figure 9. We improved Figures 4 and 10.
- "The idea of application of Evolutionary Algorithm should be explained in a more detailed way (application of mutation operator etc.). Some brief recalling of the Tiny GP algorithm should be given. More clear explanation (including a graphical illustration) of the adaptation of this algorithm to the problems under consideration should make the paper more practical."
- Response: We detailed the explanation of Evolutionary Algorithms and provided an explanation on how TIny GP works (see sub-section 4.4).
- Response: We detailed the explanation of Evolutionary Algorithms and provided an explanation on how TIny GP works (see sub-section 4.4).
- " Editorial form of the paper must be improved."
- Response: There was an uniformization of tables style presented in sub-section 2.2.
- Response: There was an uniformization of tables style presented in sub-section 2.2.
English language and style corrections have been addressed by revising the written text in the document. They can be found through the whole document.
* All the revisions performed to the article have been highlighted in green so that reviewers can easily track the changes.
Reviewer 3 Report
This article introduces an evolutionary framework for auditory-visual relationships and its findings are relevant to better understand the foundations of auditory-visual associations mediated by emotions and perception.
The authors argue that crossing perceptual and emotional findings with auditory and visual characteristics may bring additional value to the expression of cross-modal associations of a creative nature.
The main contributions of this article lie in bringing together diverse research domains such as music, image, emotion, cross-modal associations and using evolutionary computation to abstract auditory-visual associations.
Authors highlight three key benefits (i) Offering a systematic and multidisciplinary approach to bridge the domains of music and image based on a shared vocabulary of perceived emotions, (ii) providing a test hypothesis for a system that allows the exploration of auditory-visual relationships, and (iii) contributing to a better understanding to the transposition of human creative processes in computational approaches.
This article shows a well-organized structure and is written in a technically correct manner and introduces a relevant novelty addressing auditory-visual relationships and using genetic programming techniques to evolve audio-visual dialogues.
Furthermore, this article gives enough and relevant reference to prior work and the quality of experimental results are presented in two different phases.
In the first analysis phase, authors started by counting how many times each parameter is associated with a specific emotion. In the second analysis phase, authors added the number of times each property is associated with each emotion.
It could be interesting showing a graphical representation of the emotional associations with visual and musical properties. Furthermore, a statistical analysis could be useful to show the corresponding distributions.
In figure 1, the graphical representation of the framework could be improved to help the reader having a better understanding, it might be good idea using colors in it.
Following a list of comments to improve mainly the figures:
In figure 2, it will improve the caption explanation mentioning that it shows a kind of graphical analysis of the Kandinsky painting.
In Figure 3, the authors show a visual interpretation of the properties and values described above, but it is not clear whether this representation is an entirely new proposal by the authors.
In Figure 4, it will be helpful adding an explanation about how the images are generated and maybe a construction example could be helpful too.
Author Response
- "It could be interesting showing a graphical representation of the emotional associations with visual and musical properties. Furthermore, a statistical analysis could be useful to show the corresponding distributions."
- Response: Addressing this comment was challenging as we consider Table 1 to be a representation of the emotional associations with visual and musical properties. However, we have added a figure (Figure 3), as an example on how these relationships can be illustrated. Regarding the statistical analysis proposed, we would need more time to revise the article and perhaps a clarification on the comment to address it better. We have for instance added a standard deviation metric to the quantitive analysis in experimental results section (see Table 9 in section 5, page 23).
-
"In figure 1, the graphical representation of the framework could be improved to help the reader having a better understanding, it might be good idea using colors in it."
- Response: Figure 1 is now Figure 4 in the article. We have improved the scheme by adding more details to the process and placed each module vertically to add a better definition to the image (page 10).
- "In figure 2, it will improve the caption explanation mentioning that it shows a kind of graphical analysis of the Kandinsky painting."
- Response: Figure 2 is now Figure 6 in the article. We improved both the image lines contrast and added to the caption that it is an analysis of a Kandinsky painting (page 12).
-
"In Figure 3, the authors show a visual interpretation of the properties and values described above, but it is not clear whether this representation is an entirely new proposal by the authors."
- Response: Figure 3 is now Figure 8 in the article. This figure has been improved by adding examples on the grammar rules and it has been clarified that this is a new proposal by the authors (page 16).
- "In Figure 4, it will be helpful adding an explanation about how the images are generated and maybe a construction example could be helpful too."
- Response: Figure 4 is now Figure 9 in the document. We have added an explanatory text detailing the construction behind the images in Figure 9 (page 16). Additionally, we introduced a scheme in Figure 9 illustrating the generation process (page 17).
- Response: Figure 4 is now Figure 9 in the document. We have added an explanatory text detailing the construction behind the images in Figure 9 (page 16). Additionally, we introduced a scheme in Figure 9 illustrating the generation process (page 17).
English language and style corrections have been addressed by revising the written text in the document.
* All the revisions performed to the article have been highlighted in green so that reviewers can easily track the changes.
Round 2
Reviewer 2 Report
The Authors addressed all my concerns and comments in a satisfactory way.
Therefore, now I recommend the paper for publication.
This is a very interesting research direction!